# SugarDrawer: A Web-Based Database Search Tool with Editing Glycan Structures

**DOI:** 10.3390/molecules26237149

**Published:** 2021-11-25

**Authors:** Shinichiro Tsuchiya, Masaaki Matsubara, Kiyoko F. Aoki-Kinoshita, Issaku Yamada

**Affiliations:** 1The Noguchi Institute, Tokyo 173-0003, Japan; tsuchiya@noguchi.or.jp (S.T.); matsubara@noguchi.or.jp (M.M.); 2Graduate School of Science and Engineering, Soka University, Tokyo 192-8577, Japan; kkiyoko@soka.ac.jp; 3Glycan & Life Systems Integration Center (GaLSIC), Soka University, Tokyo 192-8577, Japan

**Keywords:** glycan structure, SNFG notation, database search, drawing software, WURCS, GlycoCT

## Abstract

In life science fields, database integration is progressing and contributing to collaboration between different research fields, including the glycosciences. The integration of glycan databases has greatly progressed collaboration worldwide with the development of the international glycan structure repository, GlyTouCan. This trend has increased the need for a tool by which researchers in various fields can easily search glycan structures from integrated databases. We have developed a web-based glycan structure search tool, SugarDrawer, which supports the depiction of glycans including ambiguity, such as glycan fragments which contain underdetermined linkages, and a database search for glycans drawn on the canvas. This tool provides an easy editing feature for various glycan structures in just a few steps using template structures and pop-up windows which allow users to select specific information for each structure element. This tool has a unique feature for selecting possible attachment sites, which is defined in the Symbol Nomenclature for Glycans (SNFG). In addition, this tool can input and output glycans in WURCS and GlycoCT formats, which are the most commonly-used text formats for glycan structures.

## 1. Introduction

Database integration using Linked Open Data technologies [1] has been promoted in many science fields. In the field of life sciences, integration is progressing and contributing to collaboration between different research fields, such as proteomics and genomics, to solve cross-cutting issues across fields [2]. The importance of database integration is recognized in the glycosciences as well. After the international glycan repository GlyTouCan was released in 2015 [3], integration greatly progressed worldwide. Today, GlyTouCan has over 120,000 entries registered by researchers worldwide. This number is expected to increase in the future because of the release of the GlyCosmos glycoscience portal [4], which provides an integrated interface for glycoscience data, and is increasingly collaborating with other research fields to integrate glycoconjugate data, such as glycoproteins and glycolipids. We are also collaborating with the worldwide Protein Data Bank (wwPDB), which recently performed an update of their data representation for carbohydrates, as a result of consultation with the glycoscience community (Carbohydrate Remediation) [5]. This remediation provides access to glycan 3D structural information in the PDB in a more Findable, Accessible, Interoperable and Reusable (FAIR) manner. To facilitate database access to the increasing number of glycan entries in GlyTouCan and GlyCosmos, we have developed infrastructure to access glycan related information more efficiently, e.g., we provide a web Application Programming Interface (API) for obtaining GlyTouCan IDs from WURCS [6], which is the main text representation used in GlyTouCan. WURCS is now also included as one of the linear descriptors of carbohydrates in the PDB.

On the other hand, many web users need a useful search system for accessing desired glycan structures. Although the web API described above can provide access to glycan entries, it is not easy for many experimental researchers to use such informatics technologies. A search system should enable non-experts to search glycans easily, and inputting and searching glycans should not take much time. Thus, we considered the following requirements to develop a tool for glycan search: (1) users should be able to input glycan structures by editing symbolic nomenclature via the tool; (2) database search should be available; (3) all processes should be fast and completed online.

The symbolic nomenclature described in requirement (1) refers to glycan structure representation with symbols for representing monosaccharides and lines for describing linkages between monosaccharides. The Symbol Nomenclature for Glycans (SNFG) [7] proposed by a wide community of glycoscientists proposes a standard nomenclature for glycans. This nomenclature covers not only fully defined glycan structures but also glycans in ambiguous states, such as glycan fragments which are those containing underdetermined linkages. Considering the fact that many glycan fragments are registered in GlyTouCan and thereby are queried frequently, the tool which we are considering must support glycan fragments as well. Regarding requirement (2), the web API, described above, can be used for database search online via this tool.

For development of the tool, we used or referenced some existing tools. GlycanBuilder [8], developed during the EuroCarb project [9], has been long used by many glycoscientists. This tool provides excellent features for editing glycan structures in symbolic nomenclature, and its features were updated in GlycanBuilder2 to make it compatible with SNFG representation as well as some special glycan structures such as cyclic glycans [10]. This tool was upgraded to run in a web environment and is used in the database search systems of GlyTouCan and UniCarbKB [11], but has some limitations and difficulties in terms of usage and maintenance as a web application. SugarSketcher [12], developed by the Swiss Institute of Bioinformatics, is a web-based glycan editing tool. This tool can edit many glycan structures, except for ones in an ambiguous state, and provides a simple editing interface for use on smartphones and tablets, but it requires many steps even to add a monosaccharide. This tool can be used as a web application because all the systems are coded in JavaScript. This tool can export GlycoCT [13], which is supported by many software tools and databases. However, because of the lack of an online database search functionality in this tool, users must take the GlycoCT outputted by this tool and then manually enter the string into the search interface for databases which provide GlycoCT search. The code is freely available from their GitHub repository at https://github.com/alodavide/sugarSketcher (accessed on 30 October 2021). GlycoGlyph [14] is another tool that is available online and provides similar features in SugarSketcher, such as GlycoCT export. This tool provides a GlyTouCan search system, which can obtain GlyTouCan IDs, and the links for not only the inputted glycans but also ones with different anomeric states can be obtained from GlyTouCan.

After consideration of all these tools, we decided to develop a new web-based glycan search tool, SugarDrawer, based on SugarSketcher functionalities, but additionally providing an updated glycan editing interface and some additional features. In this manuscript, we describe the following as the main features of our tool: (1) an intuitive editing experience similar to GlycanBuilder; (2) editing structures and exporting of text formats supporting glycan fragments; (3) GlyTouCan database access through this tool. SugarDrawer can be accessed at https://glycoinfo.gitlab.io/sugardrawer/sugar-drawer-pages/ (accessed on 30 October 2021). The code is available on GitLab at https://gitlab.com/glycoinfo/sugardrawer/sugardrawer (accessed on 30 October 2021).

## 2. Results

In this section, we will describe three features: (1) basic glycan editing via the user interface; (2) handling of glycan fragments; (3) database search for glycans inputted on the interface. Feature (2) also contains text conversion for glycan fragments.

### 2.1. The Interface for Editing Glycans

This tool provides an intuitive interface for editing glycans, based on similar functionalities in GlycanBuilder (Figure 1). Basic functionalities to create a glycan are provided by the blue buttons at the top of the interface. The “Load Structure” button provides some template structures which allow users to create their structures based on existing known structures. Once users click this button, they can select template structures from the following glycan types: N-linked glycan; O-linked glycan; glycolipid; glycosaminoglycan (GAG). The “Add Monosaccharide” button is used for adding monosaccharides to the canvas. This button displays a categorized list of monosaccharides based on SNFG. As a shortcut, frequently used monosaccharides can be selected directly from the row of symbols displayed under the blue buttons. The selected monosaccharide is added to the glycan on the canvas by clicking on the acceptor monosaccharide to which it should be attached. If there is no glycan on the canvas, the selected monosaccharide will be drawn by clicking on the blank canvas. The “Add Substituent” button provides a list of substituents which can be added to a monosaccharide. The selected substituent from the list can be added to a monosaccharide on the canvas when the monosaccharide is clicked. The “Add Fragment” button allows users to edit glycan fragments. We describe this feature in detail in Section 2.2.

The glycan created on the canvas can be edited to specify more details. Those that can be edited will be highlighted in red when the mouse hovers over them. Clicking on a red monosaccharide, substituent, or linkage will display various pop-up windows as shown in Figure 2. For example, when a highlighted substituent is clicked, a window for selecting possible linkage positions will be displayed. When a highlighted linkage is clicked, the pop-up window allows the user to select from combinations of possible anomers and linkage positions.

When a highlighted monosaccharide is right-clicked, users can select possible stereo configurations (Isomer) and ring sizes. Users can also select possible attachment sites (Connected). This feature is discussed in Section 2.2. in detail. The eraser button can delete the selected monosaccharide, and the button with two arrows can replace the selected monosaccharide with a different one.

Returning to the top menu of the SugarDrawer interface, the red buttons to the right of the blue buttons explained earlier are for “Undo”, “Redo”, “Clear”, “Help”, and “Normalize” features, respectively from left to right. The “Undo” and “Redo” features undo and redo the last editing operation on the canvas, respectively. The “Clear” feature clears all structure data on the canvas. Note that once the structure is cleared, the structure cannot be edited again even with the “Undo” and “Redo” features. The “Normalize” feature normalizes monosaccharide symbols on the canvas in accordance with SNFG rules. For example, if the glucose symbol has an N-acetyl group on carbon 2, the symbol will be normalized to the N-acetyl glucosamine symbol as shown in Figure 3.

The gray buttons along the bottom of the interface provide text conversion and database search features. Users can add glycans to the canvas from GlycoCT or WURCS text strings using the “Import String” button. Conversely, users can obtain GlycoCT and WURCS strings for the glycan on the canvas using the “Export String” button. The “Search” button is for the database search described in Section 2.3.

### 2.2. Handling Glycan Fragments

To handle glycan fragments, we defined glycan fragments as those consisting of a core and one or more fragment parts. Therefore, the core part must be drawn before adding fragment parts when drawing glycan fragments.

Users can add a monosaccharide as a fragment part by selecting a monosaccharide of the core part and then clicking the “Add Fragment” button, similar to using the “Add Monosaccharide” button. After the monosaccharide is added, the monosaccharide is drawn to the left of the core part separated by a bracket as shown in Figure 4. This representation is according to SNFG rules.

To select possible attachment sites of the fragment part, users can use the pop-up window for editing monosaccharide structure information as described above. In the pop-up window, the possible attachment sites on the core part can be selected, as shown in Figure 5. To indicate the potential connection sites of the fragment part, a unique number assigned to each fragment part is shown on the linkage of the fragment part as well as the monosaccharides of the core part. These numbers are not shown when all possible sites on the core part are selected. These fragment glycans can also be imported and exported in GlycoCT and WURCS formats using “Import String” and “Export String”, respectively (Figure 6).

### 2.3. Database Search

GlyTouCan database search is also available in SugarDrawer. Once the “Search” button is clicked, the glycan on the canvas is searched for in GlyTouCan. If found, the result is displayed under the gray buttons with the matching GlyTouCan ID and image. The GlyTouCan ID is a link to the GlyTouCan entry page (Figure 7).

## 3. Discussion

GlycanBuilder2 allows users to edit glycans using SNFG symbols on the web, and it can also be utilized as a database search tool as well as for glycan text conversion such as with WURCS and GlycoCT. It currently supports the following types of glycans: simple and branched glycans, containing substituents such as sulfated monosaccharide and glycan fragments.

Various glycan editing tools have been developed by several groups. We compared these various glycan editing tools and summarized them in Table 1. All of the glycan editing tools could support standard glycans found mainly in mammalian organisms, and all these tools could represent them using SNFG. GlycanBuilder2 and SugarDrawer can represent glycan fragments using brackets and glycan subgraphs. On the other hand, SugarDrawer provides unique editing features for glycan fragments, which allow users to select attachment sites of the fragment part on the core part. Other editing tools, such as GlycanBuilder, can handle glycan fragments but do not have such features for specifying specific attachment sites. Moreover, all functionality in SugarDrawer, including editing of the glycan fragment and database search features, are web-based.

Regarding database search, we use a web API for searching GlyTouCan IDs using WURCS. We implemented this feature such that it could be easily used by other databases as well. However, the database available in SugarDrawer is limited to the GlyCosmos. On the other hand, the “Get GlyTouCan ID” function in the GlycoGlyph obtains the accession number of the GlyTouCan from editing glycan and provides detail of this structure from GlyTouCan, GlyGen, PubChem, and ChEBI, using the GlyCosmos API. Thus, we plan to provide the details of the glycans obtained from these databases in the “Search” function of SugarDrawer in the future. Since wwPDB now provides WURCS as a linear descriptor in their data category, this “Search” function can also be used to search for glycan structures in wwPDB using WURCS. However, at the time of writing, there is no system or API for accessing WURCSs in wwPDB directly, and this is left for future work.

## 4. Materials and Methods

We chose JavaScript ES2015 (ES6) as the development language for SugarDrawer. Node.js (10.19.0) and Node package manager (6.14.1) were used to manage dependencies of multiple libraries efficiently. React (15.6.1) and Semantic-ui-react were adapted for the window design. CreateJS-EaselJS (0.8.2) was used for generating glycan images.

Our tool uses SugarSketcher as a library, but we updated some of its features, including GlycoCT conversion, 2D coordinate generation and data structures for storing glycan structure information. The updated code is available at https://gitlab.com/glycoinfo/sugardrawer/SugarSketcher2 (accessed on 30 October 2021). Webpack (3.0.0) was used for compiling the code to use React and CreateJS-EaselJS in the browser.

### 4.1. Data Structures

We created the Liaise class for managing and storing all data in this tool. This class was used for editing glycan structures, for connecting SugarSketcher data structures and for storing the editing process by users to support undo and redo functionality.

### 4.2. Glycan Images

Image generation in this tool was implemented using the Glycan, Monosaccharide, and GlycosidicLinkage classes in SugarSketcher and the Shape class in CreateJS. Text image generation used the Text class in CreateJS combined with the GlycosidicLinkage and Substitutent classes in SugarSketcher.

### 4.3. Format Conversion

In this tool, SugarSketcher was used for GlycoCT import and export. For WURCS import and export, the GlycanFormatConverter API [15], which can convert between GlycoCT and WURCS, was used.

### 4.4. Handling Glycan Fragments

Figure 8 shows an extension of the SugarSketcher data structure for glycan fragments. Since the data structure implemented by SugarSketcher for glycans, the Glycan class, could not handle glycan fragments originally, we used the Glycan class for both the core and fragment parts of glycan fragments. While the usage of the Glycan class was unchanged for the core part, we added information for linkage and attachment sites such as structured data, Linkage information and Attachment sites, for the fragment part. The Linkage information has information of linkage positions to the core part from the fragment part, and the Attachment sites have a list of monosaccharide identifications which indicate the attachment sites on the core part. All the Glycans were stored in an array, and the first element was assumed to be the core part, while the subsequent ones were considered to be fragment parts. If the array had only one element, this indicates that it is a simple glycan, and not a glycan fragment. This array is a member of the Liaise class.

Since SugarSketcher does not have GlycoCT conversion functionality for glycan fragments, we extended two classes in SugarSketcher, GlycoCTParser and GlycoCTWriter, for GlycoCT import and export as GlycoCTParserForFragment and GlycoCTWriterForFragment, respectively (Figure 9). Specifically, the GlycoCTParser output and GlycoCTWriter input can only handle a Glycan, but GlycoCTParserForFragment output and GlycoCTWriterForFragment input can handle an array of Glycans indicating a glycan fragment. Since the “ParentIDs” and “SubtreeLinkageID” lines in GlycoCT are specific information for glycan fragments (see Figure 5), parsing and writing features for the lines indicating a fragment were newly implemented in GlycoCTParserForFragment and GlycoCTWriterForFragment, respectively. WURCS import and export can also handle these glycan fragments using this functionality and the GlycanFormatConverter API.

### 4.5. Database Search

This new tool provides GlyTouCan database search functionality based on WURCS. In this system, we called a web API which receives WURCS and returns a GlyTouCan ID. We also used another API which generates glycan images using the GlyTouCan ID to show the images in the same way as GlyTouCan. The GlyTouCan ID is hyperlinked to the GlyTouCan entry page, and the generated glycan image is displayed in a table. These web APIs are provided as GlyCosmos web resources (https://glycosmos.org/glycans/show/ (accessed on 30 October 2021).

## 5. Conclusions

We developed a new web-based glycan database search and drawing tool, SugarDrawer. This tool can edit a glycan fragment and its attachment sites and draw them according to SNFG rules. Furthermore, this tool provides easy and fast operations for editing glycan structures using windows for editing specific structure information. Moreover, users can also obtain WURCS and GlycoCT, which are often used in many application tools and databases. GlyTouCan database search is also available for not only simple glycans but also glycan fragments. As the result of the database search, users can access the corresponding GlyTouCan entry page. This tool is currently utilized in the GlyCosmos portal website under Glycan Search (https://glycosmos.org/search/glycans/graphic (accessed on 30 October 2021).

## Figures and Tables

**Figure 1 molecules-26-07149-f001:**
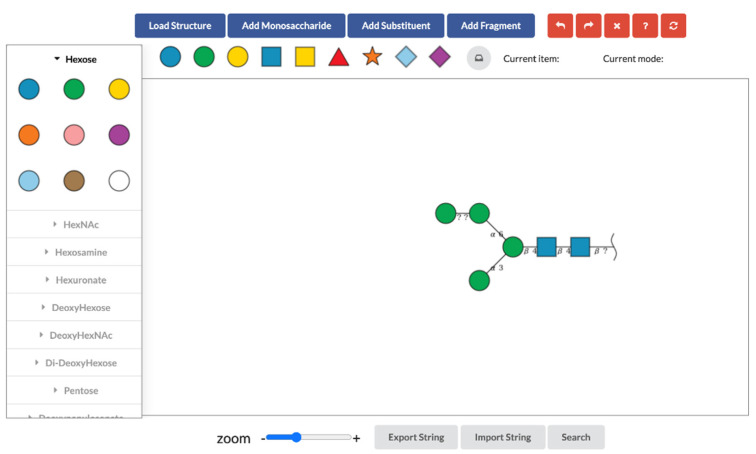
The interface of SugarDrawer.

**Figure 2 molecules-26-07149-f002:**
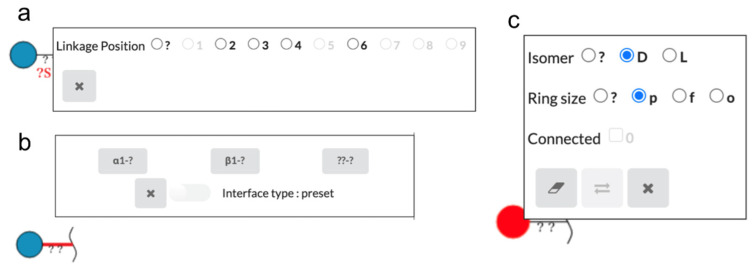
Pop-up windows for editing monosaccharides, substituents, and linkages between them: (**a**) the window to select possible linkage positions is displayed when the substituent is clicked; (**b**) the window to select combinations of possible anomer and linkage positions is displayed by clicking on the glycosidic linkage; (**c**) the window to edit the monosaccharide is displayed by right-clicking on the monosaccharide. This contains selections for stereo configurations (Isomer), ring sizes and attachment sites, as well as buttons to delete and replace the selected monosaccharide.

**Figure 3 molecules-26-07149-f003:**
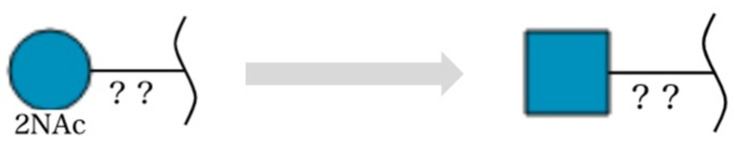
An illustration of how a glucose symbol with an N-acetyl group on carbon 2 can be normalized to the N-acetyl glucosamine symbol using the “Normalize” feature.

**Figure 4 molecules-26-07149-f004:**
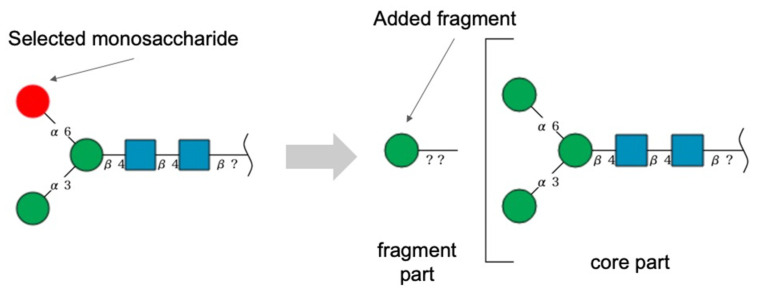
Addition of a monosaccharide as a fragment part using “Add Fragment”.

**Figure 5 molecules-26-07149-f005:**
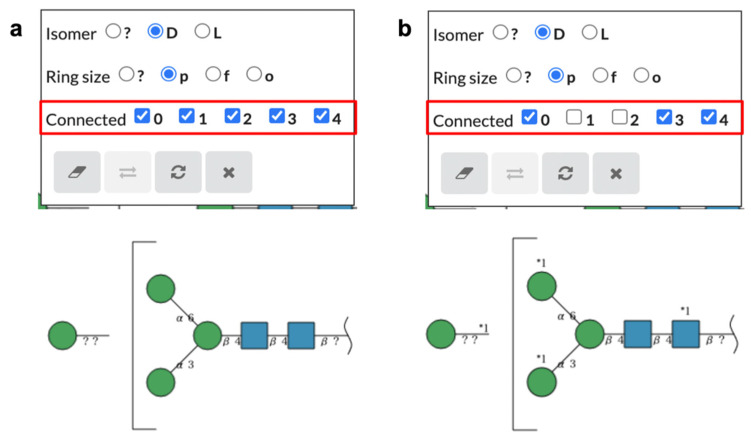
Windows for editing attachment sites of a fragment part. Users can edit attachment sites on the core part of glycan fragments by selecting the “Connected” checkbox in the pop-up window. If the attachment sites of a fragment part are specified, a number following “*” is assigned to the fragment part and the selected attachment sites are indicated by the number on the symbols of attaching monosaccharides. (**a**) A view of the window and glycan when all attachment sites are selected and (**b**) when not all attachment sites are selected.

**Figure 6 molecules-26-07149-f006:**
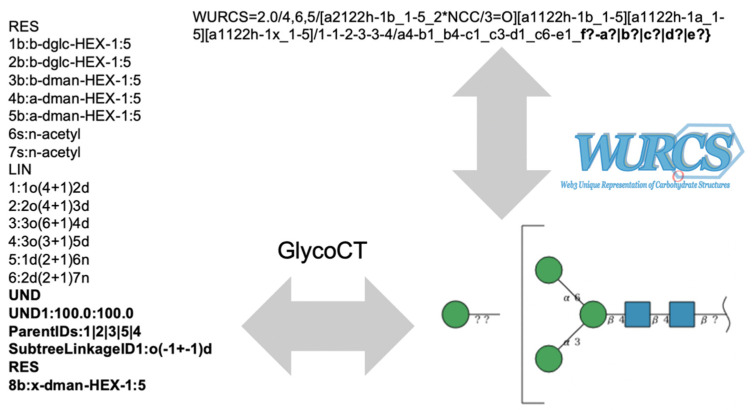
GlycoCT and WURCS containing fragment parts. The fragment parts of each format are indicated in bold.

**Figure 7 molecules-26-07149-f007:**
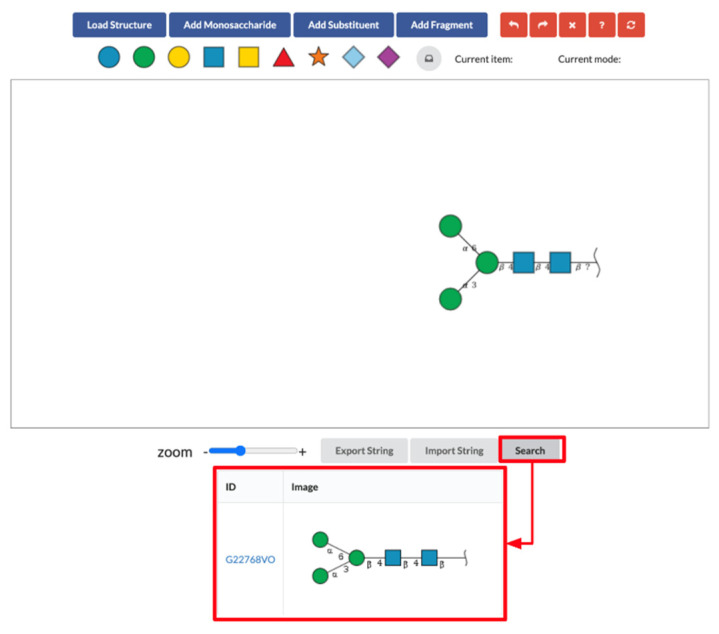
Image of glycan search. The result of the search includes GlyTouCan ID and the glycan image.

**Figure 8 molecules-26-07149-f008:**
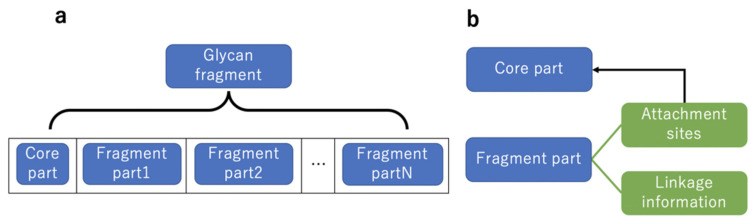
Relationships of data structures for glycan fragments. (**a**) A glycan fragment is treated as an array of data structures for core and fragment parts. The data structure for each part is defined as Glycan class in SugarSketcher. (**b**) Each Fragment part contains Attachment sites for the Core part and Linkage information.

**Figure 9 molecules-26-07149-f009:**
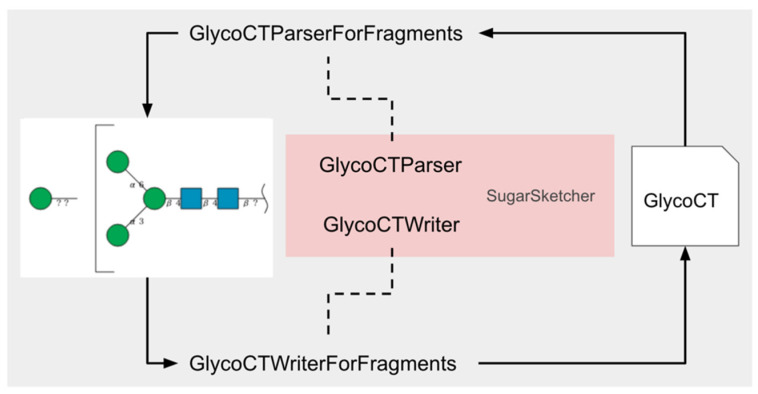
Schematic diagram of GlycoCT conversion process.

**Table 1 molecules-26-07149-t001:** Summary of features of glycan visualization tools.

	SugarDrawer	GlycanBuilder2	GlycoGlyph	SugarSketcher
Glycan editing functionality	+	+	+	+
Support SNFG	+	+	+	+
Mammalian structure	+	+	+	+
Glycan fragment	+	+		
Attachment site	+			
Output WURCS	+	+		
Database search	+		+	
Web-based	+		+	+

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
