# Peer review of "SugarDrawer: A Web-Based Database Search Tool with Editing Glycan Structures"

_molecules, 2021, doi:10.3390/molecules26237149_

Round 1

Reviewer 1 Report

This manuscript describes SugarDrawer clearly and should be accepted as it is.  It would be very much appreciated if authors provided some real examples/applications.

Author Response

    Thank you very much for providing important insights. We are grateful for the time and energy you expended on our behalf.

    As an example of the discussion in the manuscript, we have added an example text of the search function for PDB glycan structures. In addition, we have already provided an actual application of GlyTouCan's glycan search function in Figure 7. Therefore, we hope that readers can understand how to edit and search glycan structures. And we revised a few words, misspellings,  in our manuscript.

    Again, we appreciate all of your insightful comments. Thank you for taking the time and energy to help us improve the paper.

Reviewer 2 Report

Glycan is one of the important components in drug discovery, especially for biologics. They play essential roles for protein folding, oxidation, and critical for many biological studies such as pharmacokinetics and pharmacodynamics. With more than 120,000 entries, GlyTouCan is a useful database for pharmaceutical development to elucidate structures.
In this manuscript, the authors created a web-based database search tool SugarDrawer to enable experts and non-experts to easily input, edit, search and export glycan structures through GlyTouCan database on the fly. Several key features are thoroughly decribed to increase the search accuracy and efficiency. Comparing to other existing glycan visualization tools, SugarDrawer covers all of their functions, but also provide a unique feature to edit glycan fragment and its attachment sites according to SNFG rules.
Overall, the paper is clear and detailed written, and can be considered for publication. No further review is needed.

I have one suggestion for the authors. 3D structures are found to be useful to understand Glycan functions. It will be very helpful to the users if the tool can also search the input Glycan structures in RCSB or other 3D structure databases.

Author Response

    Thank you very much for providing important insights. We are grateful for the time and energy you expended on our behalf.

    As an example of the discussion in the manuscript, we have added an example text of the search function for PDB glycan structures. And we revised a few words, miss-spells,  in our manuscript.

    Again, we appreciate all of your insightful comments. Thank you for taking the time and energy to help us improve the paper.